# Temperature Changes during Implant Osteotomy Preparations in Human Cadaver Tibiae Comparing MIS^®^ Straight Drills with Densah^®^ Burs

**DOI:** 10.3390/genes13101716

**Published:** 2022-09-24

**Authors:** Nikolaos Soldatos, Huy Pham, Walid D. Fakhouri, Binh Ngo, Panagiotis Lampropoulos, Tiffany Tran, Robin Weltman

**Affiliations:** 1Department of Periodontics, School of Dentistry, Oregon Health Science University (OHSU), 2730 SW Moody Ave, Portland, OR 97201, USA; 2Department of Periodontics and Dental Hygiene, School of Dentistry, University of Texas, Health Science Center at Houston, 7500 Cambridge St., Houston, TX 77054, USA; 3Department of Diagnostic and Biomedical Sciences, School of Dentistry, University of Texas, Health Science Center at Houston, 7500 Cambridge St., Houston, TX 77054, USA; 4School of Dentistry, University of Texas, Health Science Center at Houston, 7500 Cambridge St., Houston, TX 77054, USA; 5Department of Prosthodontics, School of Dentistry, National and Kapodistrian University of Athens (NKUA), 2 Thivon St., Goudi, 11527 Athens, Greece; 6California School of Podiatric Medicine, Samuel Merritt University, Oakland, CA 94609, USA; 7Department of Clinical Sciences, School of Dental Medicine, University of Nevada Las Vegas (UNLV), Las Vegas, NV 89106, USA

**Keywords:** temperature changes, cortical bone, cancellous bone, human tibiae, dental implants, straight drills, tapered drills, dental implant osteotomy preparations

## Abstract

(1) Background: Several studies showed a sustained temperature of 47 °C or 50 °C for one minute resulted in vascular stasis and bone resorption with only limited bone regrowth over a 3–4-week healing period. The purpose of the present study was to evaluate the temperature changes (ΔΤ) that occur during the preparation of dental implant osteotomies using MIS^®^ straight drills versus Densah^®^ burs in a clockwise (cutting) drilling protocol. (2) Methods: Two hundred forty (240) osteotomies of two different systems’ drills were prepared at 6 mm depth at 800, 1000, and 1200 revolutions per minute (RPM), in fresh, unembalmed tibiae, obtained by a female cadaver. ΔΤ was calculated by subtracting the baseline temperature on the tibial surface, from the maximum temperature-inside the osteotomy (ΔT = T_max_ − T_base_). The variables were evaluated both for their individual and for their synergistic effect on ΔΤ with the use of one-, two-, three- and four-way interactions; (3) Results: An independent and a three-way interaction (drill design, drill width, and RPM) was found in all three RPM for the Densah^®^ burs and at 1000 RPM for the MIS^®^ straight drills. As Densah^®^ burs diameter increased, ΔΤ decreased. The aforementioned pattern was seen only at 1000 RPM for the MIS^®^ straight drills. The usage of drills 20 times more than the implant manufacturers’ recommendation did not significantly affect the ΔΤ. A stereoscopic examination of the specimens confirmed the findings. (4) Conclusions: The independent and synergistic effect of drills’ diameter, design and RPM had a significant effect on ΔΤ in human tibiae, which never exceeded the critical threshold of 47 °C.

## 1. Introduction

Osseointegration of dental implants within alveolar bone requires a cascade of biologic healing events resulting in the direct contact of the bone with the implant surface [1]. Multiple factors may affect healing while preparing a surgical site for implant placement [2]. Frictional heat generated from the drilling sequence is dispersed to surrounding tissues and can cause local bone necrosis and detrimentally influence the physiology of the alveolar bone [3]. Several studies showed that temperatures over 47 °C may cause bone resorption with very limited bone regrowth over the healing period [4,5,6,7]. Trisi et al. reported in a cortical bone ovine model that temperatures of 60 °C for more than one minute did not interfere with osseointegration of the placed implants but noted conical-shaped marginal bone resorption with measurable infrabony pockets [8]. This early crestal bone loss may create a non-cleansable environment and act as a catalyst for future bone loss due to peri-implantitis.

Factors that may influence the temperature changes (ΔΤ) during implant osteotomy preparations include (i) drill geometry and design, (ii) bone density and cortical thickness, (iii) drilling sequence, (iv) one step drilling or intermittent movements, (v) drill use, (vi) internal or external irrigation, and (vii) pressure applied from the operator to the handpiece [6,9,10,11,12,13,14,15,16,17,18,19].

The biologic consequences of overheating bone include protein denaturation, enzyme inactivation, osteoblast necrosis, osteoclast necrosis, and bone resorption, leading to alteration of bone-implant integration and possibly implant osseointegration failure [5,20,21,22,23].

Preliminary studies comparing straight, and tapered implant drills found significant differences in heat generation when preparing osteotomies at 800, 1000, or 1200 revolutions per minute (RPM) [24]. While neither the straight nor tapered drills produced temperatures that exceeded the 47 °C threshold, tapered drills generated significantly higher temperatures than straight drills [24]. An innovative tapered drill has been marketed to offer a “fast feed rate with minimal heat elevation” [25]. The manufacturer states that “the flutes are tipped with a proprietary chisel edge that concentrates thrust force while reducing tool chatter”. Osteotomy preparations using these drills may have less ΔΤ than conventional drilling protocols with straight or tapered designed drills.

The purpose of the present study was to evaluate the temperature changes that occur during the preparation of implant osteotomies using MIS^®^ straight drills* ^(^* MIS Implants technologies Inc., Dentsply^®^, York, PA, USA) versus the above referenced Densah^®^ burs† († Versah^®^, Jackson, MI, USA) in a clockwise (cutting) drilling protocol. The specific aims were (i) to compare ΔΤ at 800, 1000, and 1200 RPM for osteotomy preparations using a straight and the slightly tapered drill design of the Densah^®^ burs as the osteotomy width is gradually increased, based on the manufacturers’ protocol, and (ii) to observe whether either of the drill systems produce heat levels known to be conducive to thermal bone necrosis. The null hypothesis was that osteotomy preparations, with external irrigation, at 800 RPM will generate the same heat as 1000 and 1200 RPM in two different implant drill designs.

## 2. Materials and Methods

No Institutional Review Board (IRB) approval was required for the completion of the present human cadaver study. The study was funded by the Department of Periodontics and Dental Hygiene, School of Dentistry, University of Texas Health Science Center at Houston, and the implant drills were donated by two dental companies. The cadaver was donated for clinical and research purposes to the Department of Neurobiology and Anatomy, McGovern Medical School, University of Texas, Health Science Center at Houston. The relatives signed all the appropriate informed consents, and the cadaver was examined through blood testing to ensure the safety of the present study. The methodology was reviewed and approved by an independent statistician.

A seventy-five-year-old (75) female deceased patient was received from the UTHealth McGovern Medical School morgue. The patient had passed away at 8:22 am on 19 June 2020, due to complications from Lewy body dementia. The patient had no known history of osteoporosis, bone diseases, HIV, hepatitis, or cancer. At the time of death, she was 5′6″ and 145 lbs. The cadaver was immediately placed in the freezer that same day. The lower extremities were removed from the freezer (5 °F) and placed in the cooler (42 °F) to thaw on 25 June 2021. The study was performed on 28 June 2021.

An innovative translational model using human cadaver tibiae, previously developed by one of the authors (N.S.), and used in a previous study, was used in the present study [24]. Human tibiae and mandibular bone, although having different origins, possess similar compressive strength and elastic modulus [26].

Calibration: Calibration of the examiners was completed at the start of the study on a bone block analog maintained in the range from 95.2 °F to 99.6 °F (35.1–37.5 °C). Both examiners (N.S. and H.P.) took turns preparing osteotomies in a type II bone block analog (Sawbones^®^, Vashon Island, WA, USA) using a pilot drill separate from the study. Temperature was recorded before and after each osteotomy. This process was repeated until 8 consecutive ΔT measurements fell within 2 °C of one another and two the examiners’ average ΔT was within 1 °C.

Two six-inch long unembalmed tibial sections were harvested bilaterally (Figure 1).

A temperature regulated water bath, filled with sterile saline, was maintained at a range of 95.2 °F to 99.6 °F (35.1–37.5 °C). Temperature of the saline bath and osteotomies were recorded with an oral thermometer (REF MDS9950) and K-type thermocouple (Fisher Scientific^®^, Hampton, NH, USA 15-078-187, range −58 to 2000 °F, resolution 0.1°/1°, sampling rate 2.5 times per second), with an ultra-fast response naked bead probe (maximum range 260 °C), respectively (Figure 2) [24]. The room temperature was kept at a constant 68 ± 1°F (20 °C).

To prepare the osteotomies, the tibial sections were removed from the water bath and placed on a countertop for drilling. After 2–3 osteotomies, the sections were returned to the water bath to maintain the temperature as close to human body temperature within the bounds of study protocol. Repeated measurements of the tibial bone thickness were taken. The study designed the osteotomy depth at 6 mm (3 mm cortical and 3 mm cancellous bone) (Figure 3).

Six experimental groups were utilized. Both Densah^®^ burs and MIS^®^ straight drills were employed to create implant osteotomies each at 800, 1000, and 1200 RPM in a clockwise rotation, with external irrigation with sterile 0.9% sodium chloride saline to accommodate a 5.0 × 6.0 mm tapered implant (SEVEN^®^, MIS Implants technologies Inc., Dentsply^®^, York, PA, USA). For both systems, the manufacturers’ recommended protocols and drills were used. In addition to the 20 recommended osteotomies from both manufacturers, 20 more osteotomies were performed with the same drills. For the straight drill† protocol, the osteotomy was sequentially enlarged using 2.4 mm, 2.8 mm, 3.2 mm, and 4.0 mm diameter drills (Figure 4). For the tapered drills*, the sequence was 2.3 mm, 4.0 mm, 4.3 mm, and 4.5 mm average diameter drills (Figure 5).

The same spike bur (1.6 mm) was used in both systems to initiate the osteotomies. Each osteotomy was spaced apart by at least 2 mm of bone accounting for the respective width of the final drill and placed at least 2 mm from the edge of the tibial sections. Consecutive osteotomies were performed on opposite ends of the tibial sections to allow the dispersed heat to dissipate before another osteotomy was performed. Between drills, the tibial sections were returned to the water bath to best replicate internal body temperatures. A baseline measurement was recorded on the osseous surface prior to preparation. The probe was then inserted into the prepared osteotomies’ walls immediately following the osteotomy preparation with each consecutive drill (Steps 1–3, Figure 6, Figure 7 and Figure 8).

This process was repeated following the sequential surgical protocol detailed above [24]. All values were recorded on an Excel^®^ spreadsheet for statistical analyses. ΔΤ was calculated by subtracting the baseline temperature from the maximum temperature recorded immediately after drilling for each drill diameter (ΔT = T_max_ − T_baseline_).

Statistical analyses: For the statistical analyses, the R statistical software was used (R Core Team 2017) [27]. A generalized linear model (GLM) analysis was performed, specifying a gamma distribution, to assess the effect on ΔΤ using four variables: (i) drill design, (ii) drill diameter, (iii) drill usage (drill-frequency;” drillfreq”), and (iv) RPM. The variables were evaluated both for their individual and for their synergistic effect on ΔΤ with the use of one-, two-, three-, and four-way interactions [27].

## 3. Results

Figure 9 illustrates the results of the study; the Y-axis shows the ΔΤ and the X-axis shows the drill width. At all RPM, as drill diameter increased in Densah^®^ burs, ΔΤ decreased. The aforementioned pattern was seen only at 1000 RPM for the MIS^®^ straight drills. At 800 and 1200 RPM, MIS^®^ straight drills showed no significant change/effect in ΔΤ. The 95% confidence interval did show that in both drill designs, ΔΤ did not exceed the critical threshold of 47 °C, preventing the potential of producing thermal damage to biologic tissues.

Interactions between implant drill design, RPM, and drill diameter were significantly affecting ΔΤ during osteotomy preparation for both the MIS^®^ drills and Densah^®^ burs. More specifically, a three-way interaction was found in all three different RPM for the Densah^®^ burs and 1000 RPM for the straight drills (Table 1).

After completing the manufacturer recommended 20 osteotomies per drill, the drills were then continued in use to 40 osteotomies. The data were re-analyzed using drill usage (denoted as “drillfreq”) as a quantitative variable (Table 2). A four-way interaction between ΔΤ and drill design, drill width, RPM and drill usage was not found in both MIS^®^ straight drills or Densah^®^ burs. All interactions that included drill usage found no significant meaning within the limit of our study. The usage of both drill systems up to 40 osteotomies did not have a significant impact on frictional heat generation.

Stereoscopy imaging: A separate tibial section, with 3 mm cortical and 3 mm cancellous bone, was used for the preparation of three osteotomies: (i) Densah^®^-counterclockwise (osseodensification mode), (ii) MIS^®^-clockwise, and (iii) Densah^®^-clockwise (cutting mode). The tibial section with the three osteotomies was submerged in sterile water before taking stereomicroscopic images. The submerged sections were then placed under the objective lens of a Nikon^®^ Stereomicroscope (SMZ800) with a motorized stage and external light beams and adjusted to obtain good resolution images of the osteotomies in the tibial section. Several images were taken for each osteotomy at 40× magnifications. Both MIS^®^ and Densah^®^ cutting mode preparations showed similar irregularities over the osteotomy walls, suggesting non-condensed cortical bone (Figure 10 and Figure 11).

On the contrary, the osteotomy walls of the Densah^®^ counterclockwise preparation showed condensed and densified bone, similar to what Versah^®^ suggests as a finding and as the main purpose of the counterclockwise preparation (Figure 12).

## 4. Discussion

Many studies with varying methodologies addressed the problem of heat production during implant osteotomy preparation [9,15,24,28,29,30,31,32,33,34,35,36,37,38,39,40,41,42,43,44]. Trisi et al. described that bone temperature of 60 °C for 1 min during implant osteotomy preparations in an iliac crest sheep model, significantly reduced bone to implant contact [30]. Thus, controlling the variables which generate heat may enhance osseointegration and implant success. Use of different bone models, temperature measuring devices, RPM variations, drill designs, and location of temperature capture, may affect the accuracy of the temperature measurements and the comparability between studies [9,15,28,29,30,31,32,33,34,35,36,37,38,39,40,41,42,43,44]. In a rabbit tibial model, Dos Santos et al. evaluated the bone heating drill deformation and drill roughness after the preparations of implant osteotomies, using a guided and a conventional drilling protocol [28]. The guided protocol showed higher temperature, which was increased with the number of times the drills were used. Similar findings were seen for the drill roughness and deformation after 40 osteotomies, an opposite finding from the present study [28]. In a bovine rib model, Barrak et al. supported the use of a metal sleeve on a surgical guide, as an important factor for heat generation up to 2000 RPM [32]. An interesting finding was that the drill wear was notable after 210 osteotomies at 800 RPM, 120 osteotomies at 1200 RPM, and 90 osteotomies at 1500 RPM [32]. Another finding, which is in accordance with our study, is from Kirstein et al. [33]. Using three different implant systems, the highest temperature was noted after the use of the pilot drill, a finding that was seen in the present study, since the spike and pilot drills showed higher temperature than the other drills [33].

Infrared thermography is highly accurate; however, recordings through a liquid medium, such as with irrigation, could lead to inaccuracies in temperature measurements. For this reason, the thermocouple measuring unit was utilized in the present study [45,46,47]. Rashad et al. evaluated in vitro the temperature changes using two different ultrasonic devices and a conventional protocol for implant site preparations [46]. The temperature was measured 1 mm away from the osteotomy preparation site. The two ultrasonic devices significantly increased the temperature compared to the conventional protocol. Critical temperatures, over 47 °C, were found mostly on cancellous bone during the use of the ultrasonic devices [46]. Finally, the duration for the implant osteotomy preparation was significantly higher when the ultrasonic devices were used [46].

While there are multiple reasons why an implant may experience early bone loss or failure, the present study focused on temperature changes during implant osteotomy preparations and the variables affecting temperature changes. The purpose was to evaluate the temperature changes that occur during the preparation of implant osteotomies using straight drills versus a slightly tapered bur design in a clockwise (cutting) drilling protocol for simulation of the placement of a 5.0 mm diameter, 6.0 mm length (MIS SEVEN^®^) implant. The study used a highly translational human cadaver tibial model, previously developed by one of the authors (NS). This study goes beyond previous research by comparing multiple variables at one time in one-, two-, three- and four-way interactions, which is similar to what would be seen in clinical practice. The tibial model consisted of both 3 mm cortical and 3 mm cancellous bone. In cancellous bone, there are more blood vessels and less trabeculae filled by a network of rod- and plate-like elements compared with cortical bone. The anatomical variance between cortical and cancellous bone allows for different responses to heat dispersal during implant osteotomy preparations [30]. Specifically, the difference in blood flow and heat sensibility between cortical and cancellous bone has a significant influence on the healing response, affecting the cancellous bone [8,48,49]. Finally, alveolar bone is anisotropic, and the porosity differs between cortical (3.5%) and cancellous (79.3%) [50].

The null hypothesis was rejected as a statistically significant difference was found between the two tested implant systems. A three-way interaction discovered between the dependent variable, ΔΤ, and the independent variables, drill design, drill diameter, and RPM. A clear pattern appeared for the Densah^®^ burs at all RPM; the measured temperature change lessened in magnitude as the width of the drills increased. Thus, the design and the width of the burs significantly influenced the ΔΤ in every RPM. For the MIS^®^ straight drills, a significant influence on ΔΤ was only found at 1000 RPM. However, with the use of external irrigation, all drill designs, diameters, and RPM produced temperatures that did not exceed the critical temperature threshold of 47 °C. The low temperatures observed may be attributed to the 6 mm depth of the implant site osteotomies. If the sites were to be prepared for longer implants, the positive cooling effect of irrigation may not have penetrated to the deeper depths, increasing frictional heat generation. The placement of the thermocouple probe inside the osteotomy added to the accuracy of the measurement of the temperature in the present study compared to the studies, where the temperature was measured 1 or more mm away from the osteotomy.

Focusing on drill design, Scarano et al. suggested that the geometry and number of flutes in the tapered drills influence the bone temperature [44]. Scarano et al. compared temperature change in a cortical bovine bone model after osteotomy preparation with a 3.7 mm triple twist cylinder drill or a quadruple twist conical 3.7 mm diameter drill. The quadruple twist conical drill generates less heat than the triple twist cylinder drill. In a similar bovine cortical model, Cordioli et al. reported significantly lower temperature recordings after osteotomy preparations with tri-flute shaped drills, when compared to twist drills [38]. The authors posited that the tri-flute drill design allows for a better cutting efficiency, The entire length of the tri-flute drill interacts with the bone distributing heat over a greater surface area. In a bovine femoral bone model at 2500 RPM with external irrigation, Chacon et al. measured the difference in heat generated by three straight design drills with sequential drilling up to 4–4.2 mm diameters. Two systems used a relief angle near the tip of the drill, while one did not. Only the drill design without a relief angle yielded a bone temperature above 47 °C [9]. Soldatos et al. prepared osteotomies in a fresh human cadaveric model. They reported that temperature changes were significantly related to the drill diameter and whether the design was tapered or straight. Tapered drills caused significantly greater heat production compared to straight drills, which never exceeded the critical threshold of 47 °C [24]. Each implant system incorporates unique variations to their drill designs, which translates to differences in heat generation during osteotomy preparation.

In the present study, the greatest heat production was found after osteotomy preparations with the spike and the initial (pilot) drill. As the osteotomies are sequentially enlarged with wider diameter drills, the magnitude of temperature changes was reduced. This trend was most notably found with the conical designed drills. Strbac et al. utilized an artificially manufactured bone specimen of bovine origin providing a homogeneous cortical (3 mm) and cancellous (7–13 mm) area. Similar to the present study, temperature changes were measured sequentially from a 2 mm diameter pilot/twist drill to the final 5 mm diameter conical drill, under differing irrigation methods. The 2.0 mm diameter initial twist drill yielded the highest temperature, with reduced heat generation recorded as the diameter of the conical drills increased, similar to the findings for the Densah^®^ burs in the present study [36]. Soldatos et al. used a similar methodology to the present study to test three different straight drills and found that only the pilot drills showed increased temperature, which exceeded the 47 °C threshold [37]. The trends in temperature change showed the same pattern as the Densah^®^ burs used in the present study [24].

Similar to the many studies reviewed in this discussion, a primary shortcoming of this current study is its in vitro nature. One of the barriers to performing the current study in an in vivo model was the sterilization of the thermocouple probe. Another potential confounding variable was that the cadaver had previously been frozen. While the cadaver was frozen quickly after death to avoid degradation, the freezing and thawing process may influence the thermo-physical characteristics of the bone. Lastly, the potential for drill “fatigue” with multiple uses may influence the cutting efficiency of the drill. Manufacturers suggest a varying number of usages per drill to maintain cutting efficiency (and thus, to limit heat generation). The present study used each drill 40 times, similar to Dos Santos et al., although not statistically significant or clinical differences were found. Batista-Mendes et al. and Koo et al. did not find any statistically significant differences either using the drills up to 40 and 50 times, respectively [31,34]. These findings are in accordance with other studies, which used the drills for 50 osteotomies without significant elevation of the temperature [31,51,52]. Sharawy et al. evaluated in an in vitro study, the temperature, the time of drilling and the time needed for the pig jawbone to return to baseline temperature after implant osteotomy preparations using various RPM. The authors suggested that the higher the RPM, the less heat generated, a finding which was not seen in the present study [53]. Additionally, the higher the RPM, the lower the duration recorded for the preparation of the implant osteotomies [53]. A recent study by Salomo-Coil et al. using polyurethane bone blocks concluded that higher RPM with external irrigation and lower RPM without irrigation were two effective methods to avoid heat generation [54]. Despite the different protocols applied, both Sharawy et al. and Salomo-Coil et al. recommended the drilling during implant osteotomy preparations to be intermittent in order to allow for the irrigation to access the entire length of the osteotomy and to provide the necessary cooling [53,54]. Finally, a randomized controlled clinical trial by Pellicer-Chover et al. agrees with Salomo-Coil et al. that the lower RPM without irrigation used during implant osteotomy preparations compared with higher RPM and external irrigation presented with similar peri-implant bone loss at 12 months of follow-up [55].

Bratu et al. and Mihali et al. suggested the use of a short drilling implant protocol (sdip) during implant osteotomy preparations, supporting the significantly reduced duration [56,57]. The sdip consisted only of the use of the pilot drill and the final drill. Compared to the conventional protocol, the sdip did not show any statistically significant difference in temperature increase and drilling torque [56]. In addition, there was no statistically significant difference on bone remodeling between the conventional and sdip 12 months after epicrestal implant placement, both showing around 1 mm of crestal bone loss [57].

This is the first study comparing Densah^®^ burs with MIS^®^ straight drills in cutting (clockwise) mode. The current in vitro model cannot account for factors such as patient blood and salivary flow, and real-time in vivo intraosseous bone temperatures.

## 5. Conclusions

Within the limits of this in vitro human cadaver tibial model study, the independent and three-way interactions of drill design, diameter, and specific RPM significantly affected the change in temperature generated during osteotomy preparations in both MIS^®^ straight drills and Densah^®^ burs. Nonetheless, within the 95% confidence interval, neither the drill systems at 800, 1000, nor 1200 RPM were observed to generate a ΔΤ that would surpass the 47 °C threshold to induce cellular damage.

## Figures and Tables

**Figure 1 genes-13-01716-f001:**
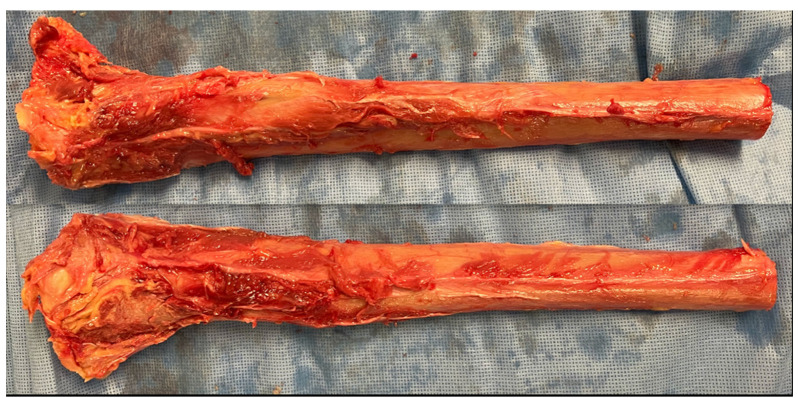
Two six-inch-long unembalmed tibial sections were harvested. Subsequently, they were placed into a 37 °C water bath, to simulate normal body temperature.

**Figure 2 genes-13-01716-f002:**
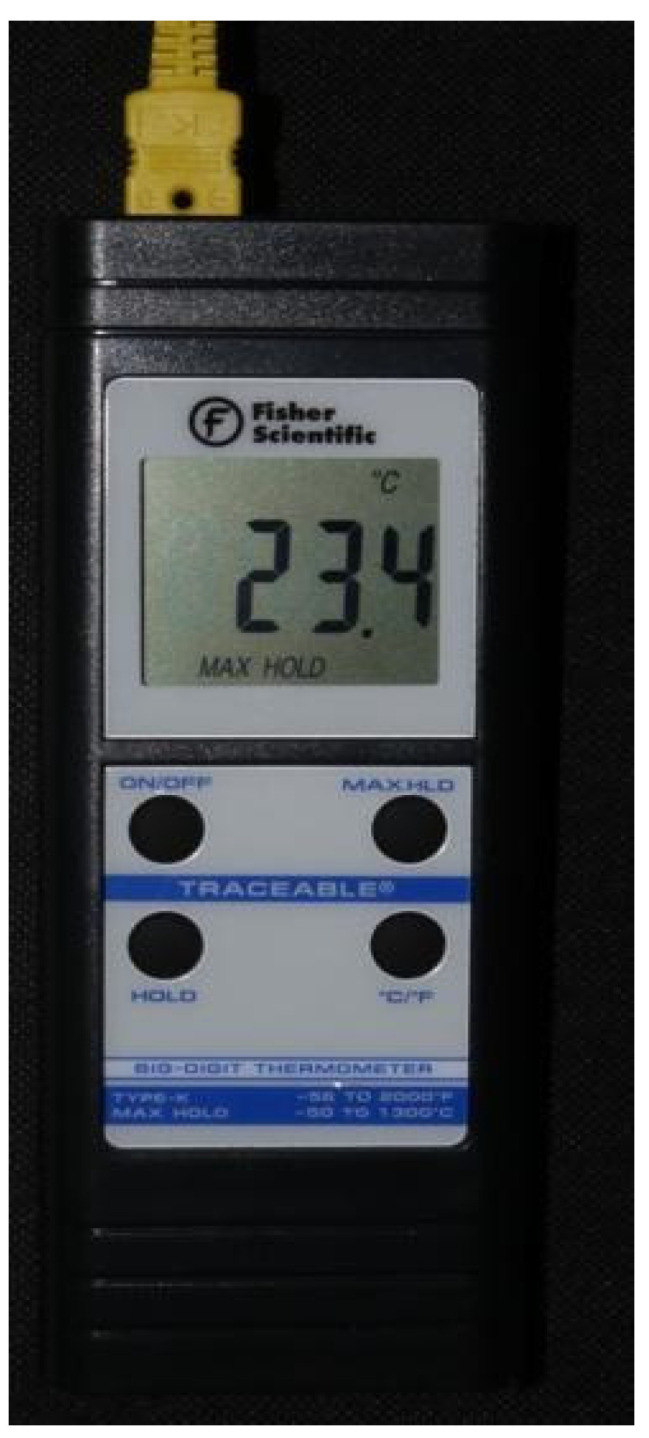
K-type thermocouple (Fisher Scientific 15-078-187, range −58 to 2000 °F, resolution 0.1°/1°, sampling rate 2.5 times per second) with an ultra-fast response naked bead probe that was used to measure the temperature.

**Figure 3 genes-13-01716-f003:**
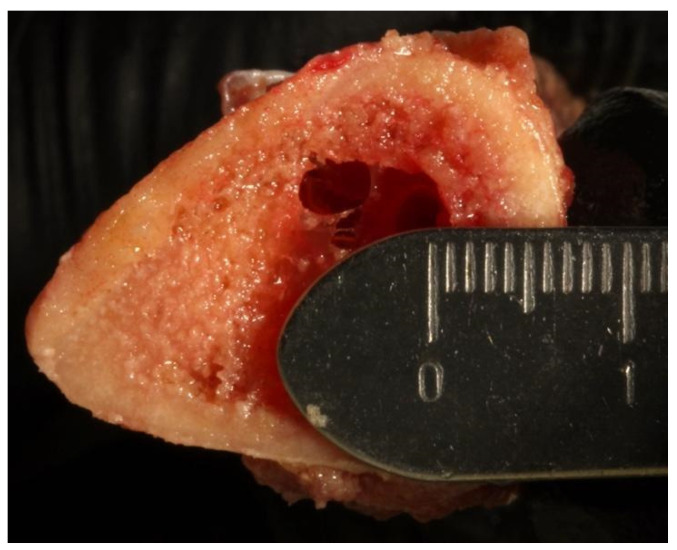
Six (6) mm thickness of the tibiae (3 mm cortical bone and 3 mm cancellous bone).

**Figure 4 genes-13-01716-f004:**
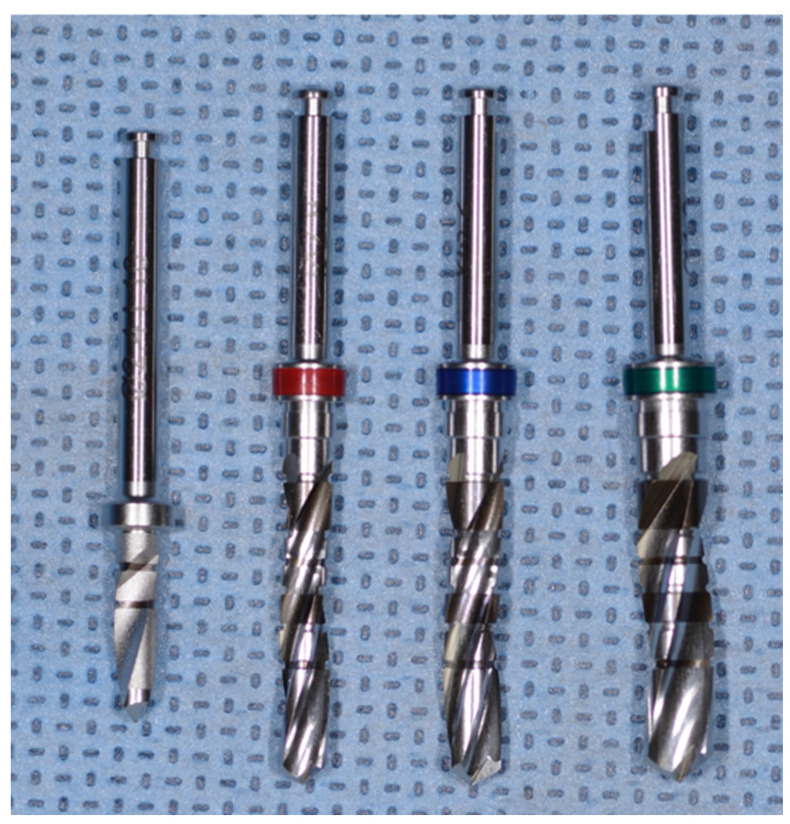
The MIS^®^ straight drills used in the study (2.4, 2.8, 3.2, and 4.0 mm.).

**Figure 5 genes-13-01716-f005:**
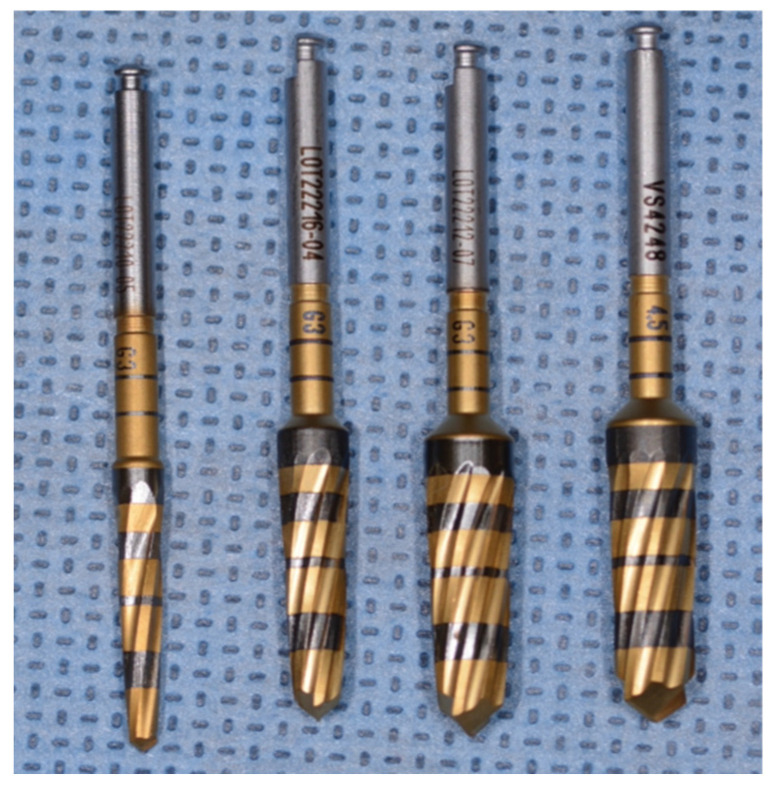
The Densah^®^ burs used in the study (2.3, 4.0, 4.3, and 4.5 mm).

**Figure 6 genes-13-01716-f006:**
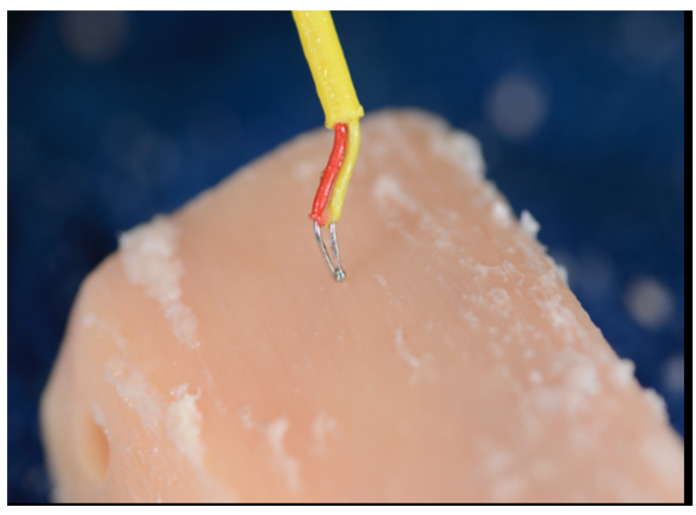
The initial temperature was recorded holding the K-Type thermocouple probe against the bony outer cortical layer (Step 1) [24].

**Figure 7 genes-13-01716-f007:**
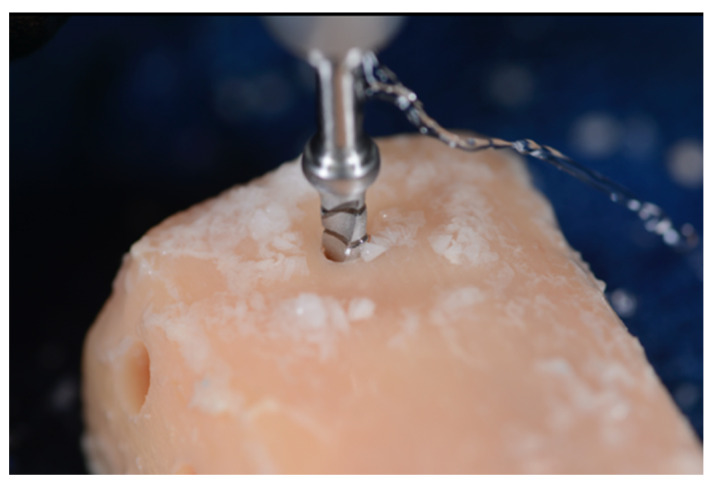
Each osteotomy was completed to the 6 mm depth, with external irrigation at 800, 1000, or 1200 RPM (Step 2) [24].

**Figure 8 genes-13-01716-f008:**
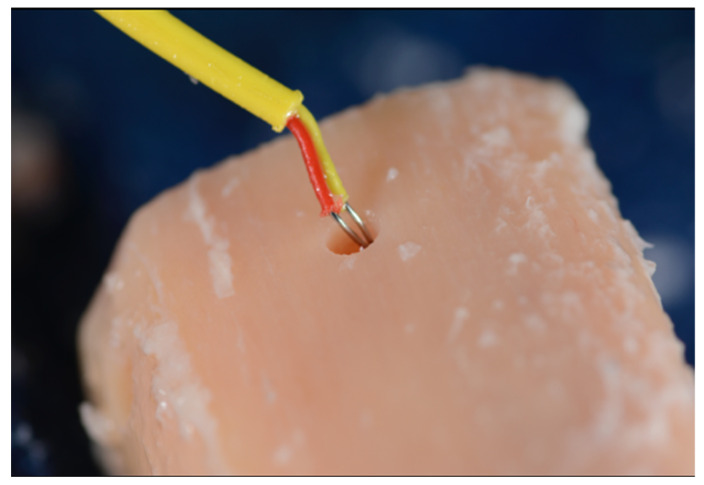
The thermocouple probe was inserted along the osteotomy wall and floor and the highest temperature value was recorded immediately after osteotomy preparation to the 6 mm depth (Step 3) [24].

**Figure 9 genes-13-01716-f009:**
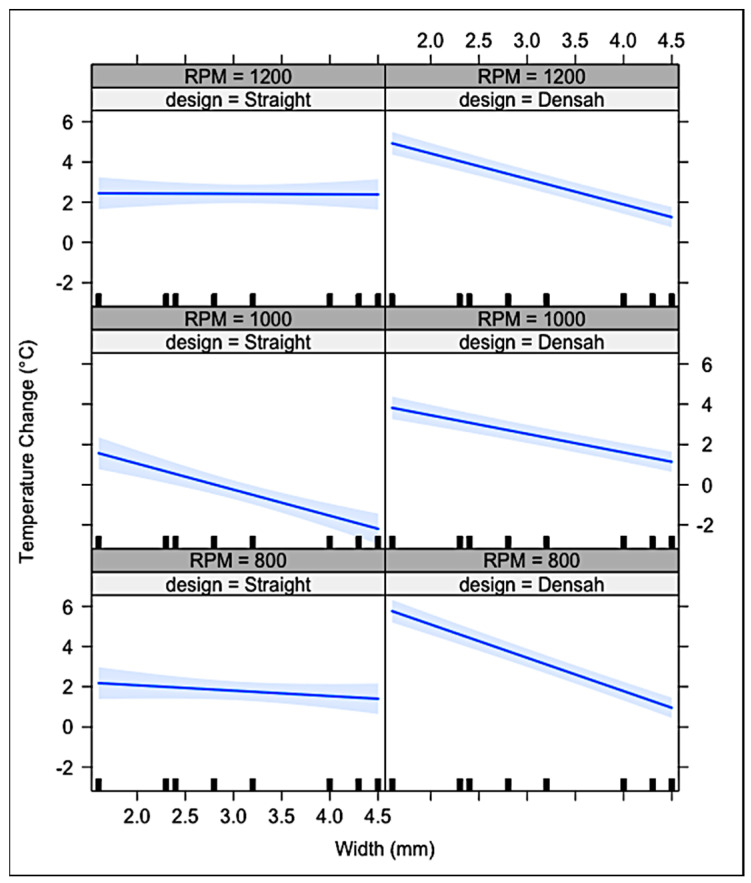
The Y-axis shows the temperature change (ΔΤ), and the X-axis shows the drill width. At all RPM, as drill diameter increased in Densah^®^ burs, ΔΤ decreased. The aforementioned pattern was seen only at 1000 RPM for the MIS^®^ straight drills. The 95% confidence interval did show that ΔΤ did not exceed the critical threshold of 47 °C in both systems.

**Figure 10 genes-13-01716-f010:**
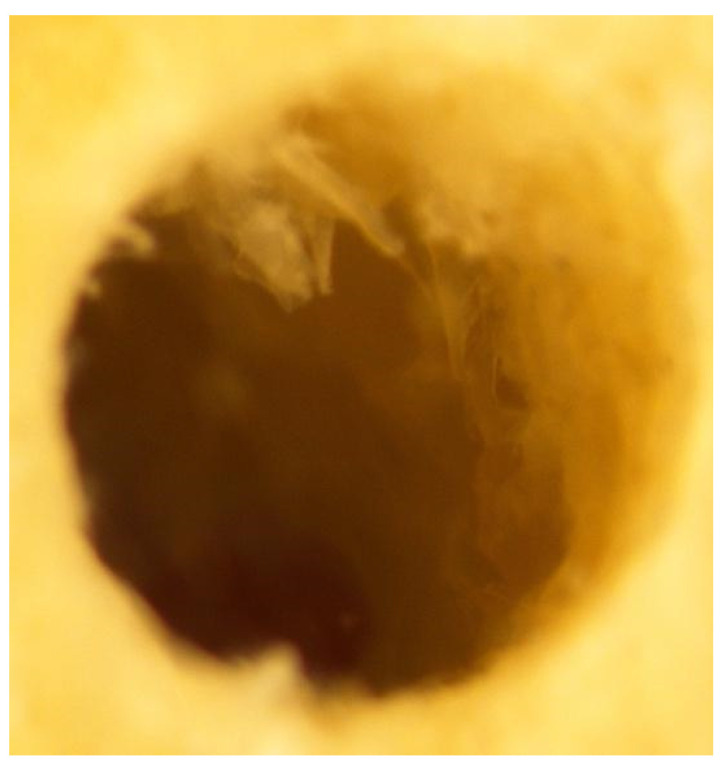
The MIS^®^ drills’ tibial section under the Nikon^®^ stereomicroscope. Irregularities can be seen over the osteotomy walls, suggesting non-condensed cortical bone.

**Figure 11 genes-13-01716-f011:**
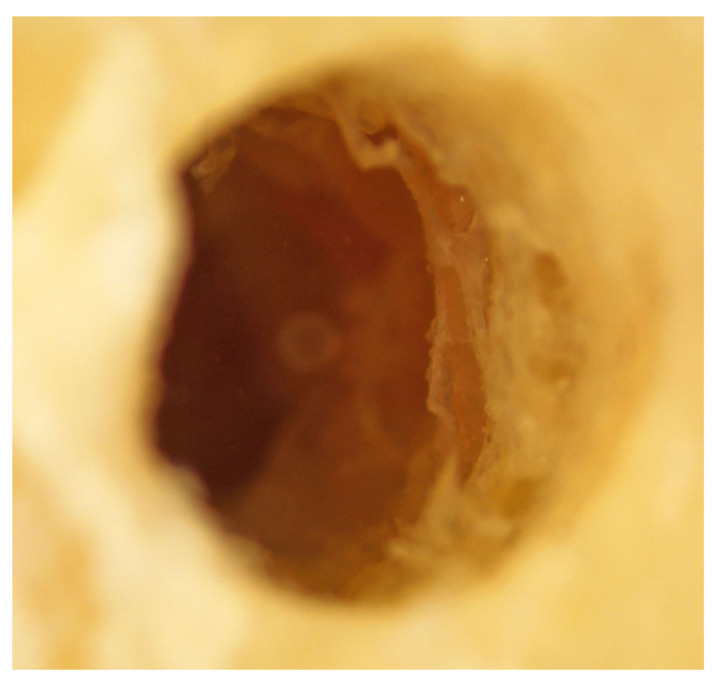
The Densah^®^ burs’ tibial section under the Nikon^®^ stereomicroscope, showing similar findings with the MIS^®^ drills’ tibial section. Irregularities over the osteotomy walls can be seen, suggesting non-condensed cortical bone.

**Figure 12 genes-13-01716-f012:**
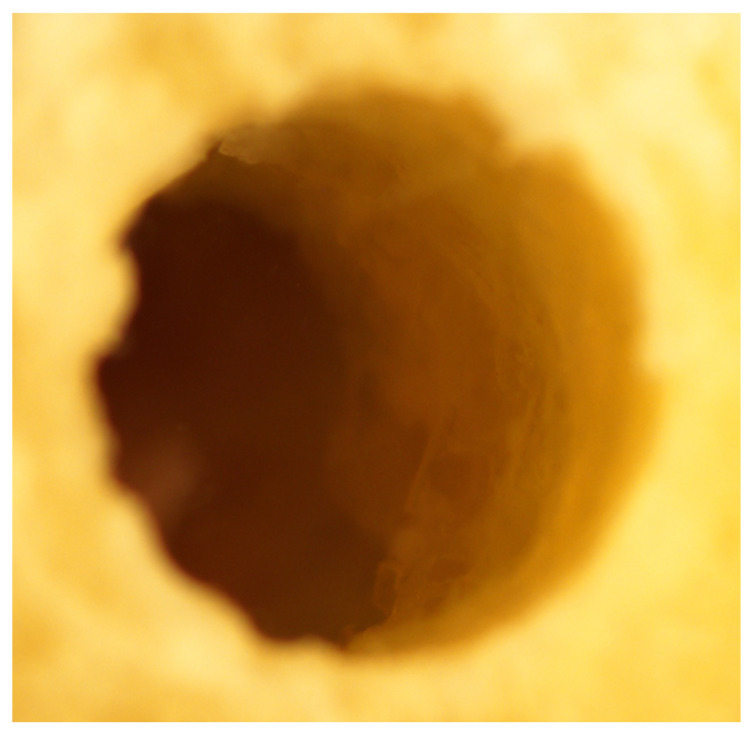
The Densah^®^ burs’ tibial section in a counterclockwise implant osteotomy preparation under the Nikon^®^ stereomicroscope, showing condensed and densified bone.

**Table 1 genes-13-01716-t001:** Analysis of deviance table; one-, two- and three-way interactions between the variables.

Source	SS	df	F	*p* Value
Design	681.5	1	160.67	<2.2 × 10^−16^ *
RPM	496.6	2	58.54	<2.2 × 10^−16^ *
Width	1312.8	1	309.51	<2.2 × 10^−16^ *
Design/RPM	228.4	2	26.93	<3.9 × 10^−12^ *
Design/width	79.2	1	18.68	<1.7 × 10^−5^ *
RPM/width	35.6	2	4.19	<0.015 *
Design/RPM/width	88.7	2	10.46	<3.17 × 10^−5^ *

* Indicates statistical significance.

**Table 2 genes-13-01716-t002:** Analysis of deviance table; one-, two-, three- and four-way interactions between the variables.

Source	F	df	DF.res	*p* Value
Drillfreq	1.82	1	227.78	<0.18
Design/drillfreq	0.11	1	230.63	<0.73
RPM/drillfreq	2.03	2	227.78	<0.13
Width/drillfreq	0.58	1	828.22	<0.45
Design/RPM/drillfreq	1.25	2	230.63	<0.29
Design/width/drillfreq	0.02	1	828.22	<0.88
RPM/width/drillfreq	2.29	2	828.22	<0.10
Design/RPM/width/drillfreq	0.10	2	828.22	<0.90

## Data Availability

Data are available upon request from the corresponding author.

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
