# Peer review of "Temperature Changes during Implant Osteotomy Preparations in Human Cadaver Tibiae Comparing MIS® Straight Drills with Densah® Burs"

_genes, 2022, doi:10.3390/genes13101716_

Round 1

Reviewer 1 Report

Regarding the drilling protocol presented in your paper, I believe you could improve the basis from which your article starts. In addition to the advantages mentioned by you, an even smaller temperature difference can appear when a shortened drilling protocol is used. This aspect will improve and strengthen the main theme related to temperature, but more than that, it will also show an option to preserve the basic principle of the article in today's situations where clinicians prefer to carry out the protocol in a much shorter time and with many more advantages. In this regard, I am attaching an article below in which you will be able to find results similar to those presented in your article. Furthermore, I recommend that you also look at certain aspects related to the diameters of the burs, which in the case of your study shows a value inversely proportional to ΔT. (1)   Link: https://pubmed.ncbi.nlm.nih.gov/29200004/   (1) Sorin G Mihali, Silvana Canjau, Anghel Cernescu, Cristina M Bortun, Hom - Lay Wang, Emanuel Bratu - "Effects of a Short Drilling Implant Protocol on Osteotomy Site Temperature and Drill Torque"     Another comparison that can be made in this sense is also represented by an example in which we have the implants inserted at the level of the dental arches (2). I specify this because compared to the previous study exemplified by me in which it was based on bovine and porcine bones, this time the similar results of these studies are also found at the level of the jaw bones. The changes made to the crestal bone level will further help to strengthen the effectiveness and highlight the results obtained in the case of your study. At the same time, comparing the drilling time performed in the case of your article with the one below will give you an advantage in terms of the situations faced by clinicians every day. The minimal remodeling that occurs at the level of the crestal bone must be specified as an advantage, as will be seen in the following article:   (2) Emanuel Bratu, Sorin Mihali, Lior Shapira, Dana Cristina Bratu, Hom-Lay Wang - "Crestal bone remodeling around implants placed using a short drilling protocol"   Link: https://pubmed.ncbi.nlm.nih.gov/25506640/

Recommended references are not mandatory to cite.

Author Response

We would like to thank the reviewer for the comments and the suggestions. We have updated the discussion section adding several studies, including the two from Bratu et al. and Mihali et al.

Reviewer 2 Report

Dear authors,

the study conducted is very interesting, as well as current. The research approach is scientifically valid and the effort to conduct the study is appreciable, avoiding the interference of confounding variables. The data collected were correctly analyzed. However, given the limitations of the in vitro study, more attention is needed to compare with that in recent literature.

For this, my advice is to:

·      Update the discussion and comparison of your results with the most updated bibliographic sources.

·      Review the English language and some typing errors.

·      Check tables and captions to facilitate the reading of the manuscript.

Best Regards

Author Response

We would like to thank the reviewer for the comments and the suggestions. We have updated the discussion section, adding several studies, we have corrected minor typing errors and we added bold text on the tables where the value indicated statistical significance.

Round 2

Reviewer 2 Report

Dear authors, thank you for the reply.
You have done a good job, but I think further improvements are needed following the method adopted so far. I consider it appropriate, in order to give greater scientific relevance to the manuscript, to compare with bibliographic sources of considerable importance. 
Furthermore, the use of bold is not enough to facilitate the reading and understanding of the data for the reader: more work needs to be done.
Best regards

Author Response

We would like to thank the reviewer for his comments. We have addressed his comments and we have added more studies in the discussion section, comparing them with the present study. At this point all the authors think that the materials and methods along with the results, are easily understandable by the readers. In case the reviewer is thinking otherwise, we would like to be more specific on his future comments. 

Kind Regards, 

Nikolaos Soldatos